# Molecular Network for Regulation of Seed Size in Plants

**DOI:** 10.3390/ijms241310666

**Published:** 2023-06-26

**Authors:** Jinghua Zhang, Xuan Zhang, Xueman Liu, Qiaofeng Pai, Yahui Wang, Xiaolin Wu

**Affiliations:** National Key Laboratory of Wheat and Maize Crop Science, College of Life Sciences, Henan Agricultural University, Zhengzhou 450046, China; zhangjinghua2016@163.com (J.Z.);

**Keywords:** seed size, IKU, ubiquitin, G protein, MAPK, transcription factor, phytohormone, miRNA

## Abstract

The size of seeds is particularly important for agricultural development, as it is a key trait that determines yield. It is controlled by the coordinated development of the integument, endosperm, and embryo. Large seeds are an important way of improving the ultimate “sink strength” of crops, providing more nutrients for early plant growth and showing certain tolerance to abiotic stresses. There are several pathways for regulating plant seed size, including the HAIKU (IKU) pathway, ubiquitin–proteasome pathway, G (Guanosine triphosphate) protein regulatory pathway, mitogen-activated protein kinase (MAPK) pathway, transcriptional regulators pathway, and phytohormone regulatory pathways including the auxin, brassinosteroid (BR), gibberellin (GA), jasmonic acid (JA), cytokinin (CK), Abscisic acid (ABA), and microRNA (miRNA) regulatory pathways. This article summarizes the seed size regulatory network and prospective ways of improving yield. We expect that it will provide a valuable reference to researchers in related fields.

## 1. Introduction

It is urgently needed to achieve increased yield for crops such as rice, corn, and wheat [1]. Seed size and number determine the final yield, and seed development also plays an important role in seed shape and weight. Seed development is coordinated by the embryo, endosperm, and integument [2], and begins with double fertilization, where one sperm cell fuses with the egg cell to produce a diploid embryo, and the other sperm cell fuses with the central cell to form a triploid endosperm [3]. The embryo and endosperm develop within the ovule, and the ovule integument eventually becomes the seed coat of the seed [4]. The endosperm plays a major role in regulating seed size [5]. Starch synthesis during endosperm development in rice affects the size of the grain [6]. Seed coat development greatly influences the final size and shape of the seed [7].

Plants regulate seed size through multiple signaling pathways [8]. Several genes that regulate seed size have been identified in different crops through various pathways, including the IKU pathway, the ubiquitin–proteasome pathway, the G protein regulatory pathway, the MAPK pathway, and the miRNA regulatory pathway (Figure 1). In addition to these well-known pathways, transcription factors and phytohormone have been shown to be involved in the control of ovule and seed size [9]. Moreover, the correct development of other plant structures is also a key determinant of grain size, and hence of crop yield, such as the spikelet hull in rice [10]. This article reviews the research results on the mechanisms that regulate seed size in plants such as Arabidopsis (*Arabidopsis thaliana* L.), wheat (*Triticum aestivum* L.), rice (*Oryza sativa* L.), and maize (*Zea mays* L.), among others.

## 2. HAIKU (IKU) Pathway

It has been reported that endosperm development is essential for seed size [2]. The IKU pathway mainly regulates early endosperm development, and includes the *IKU1*, *IKU2*, *MINISEED3* (*MINI3*), and *SHORT HYPOCOTYL UNDER BLUE1* (*SHB1*) genes (Figure 1). Wang et al. cloned the *IKU1* gene (*At2g35230*) in Arabidopsis, and the *iku1* mutant resulted in reduced endosperm growth and small seeds [11]. *IKU1* encodes a protein containing the plant-specific VQ motif. *IKU1* was expressed in early endosperm and the central cell, and the VQ motif was an essential regulatory element for seed size regulation [11]. *IKU2* (*At3g19700*) encoded a leucine-rich repeat (LRR) kinase, and overexpression of *IKU2* in Arabidopsis led to an increase in seed size, weight, and oil content [12]. *MINI3* is a WRKY10 transcription factor [13]. Mutants of *iku2* and *mini3* had a small-seeded phenotype, which was associated with reduced growth of endosperm and early cellularization [13], similar to the *iku1* phenotype [11]. *SHB1* is a gene that affects endosperm development, and normal expression of *IKU2* and *MINI3* requires *SHB1* [14]. The *shb1-D* (a gain-of-function overexpression allele) increased seed size, while *shb1* (a loss-of-function allele) decreased seed size. IKU1 interacted with MINI3, and SHB1 bound to the promoters of *IKU2* and *MINI3* to promote endosperm growth during early seed development [14]. Thus, the function of the IKU pathway in seed size control is important and could be applied in breeding for seed yield improvement [15].

## 3. Ubiquitin–Proteasome Pathway

The ubiquitin–proteasome system is the main pathway of protein degradation in cells, and includes ubiquitin, ubiquitin-activating enzyme E1, ubiquitin-binding enzyme E2, ubiquitin ligase E3, 26S proteasome, and ubiquitin-dissociating enzyme [16].

Grain width 2 (GW2), a RING-type E3 ubiquitin ligase, regulates seed development by affecting cell growth [17,18]. GhGW2-2D in cotton (*Gossypium* spp.) and AtDA2 in Arabidopsis share similar functional domains and highly conserved sequences [17]. Overexpression of *GhGW2-2D* in Arabidopsis significantly reduces seed and seedling size [17]. The ubiquitin receptor DA1 is a negative regulator of seed size [19]. In Arabidopsis, downregulation of *AtDA1* increases the sizes of seeds and organs by increasing cell division in the integuments [19]. Overexpression of *AtDA1^R358K^* in *Brassica napus* L. results in a downregulation of *BnDA1*, where the transgenic plants exhibit significantly increased biomass and seed, cotyledon, leaf, flower, and silique sizes [19]. Liu et al. identified *TaDA1* as a negative regulator of grain size in wheat [20]. Overexpression of *TaDA1* reduced grain size and weight, while downregulation of *TaDA1* using RNA interference (RNAi) had the opposite effect [20]. NGATHA-like protein (NGAL2), a transcriptional repressor encoded by the inhibitor of *da1-1* (*SOD7*), regulates seed size by limiting cell proliferation in the integuments and developing seeds [21]. Overexpression of *SOD7* significantly reduces seed size [21]. SOD7 and DEVELOPMENT-RELATED PcG TARGET IN THE APEX 4 (DPA4/NGAL3) interact with the seed size regulator KLUH (KLU) to regulate seed growth independently of DA1 [21]. GhDA1-1A in cotton and AtDA1 in Arabidopsis share highly conserved sequences and functional domains [22]. Overexpression of *GhDA1-1A^R301K^* in Arabidopsis significantly increases seed size and weight [22]. *DA2* in Arabidopsis encodes a RING-type protein with E3 ubiquitin ligase activity, which regulates seed size by limiting cell proliferation in the maternal integuments [23]. The *da2-1* mutant produces larger seeds, while overexpression of *DA2* leads to reduced seed size [23]. *DA1* and its homologous genes *DAR1* and *DAR2* are redundantly required for endoreduplication during organ development [24]. The triple mutant *da1-ko1 dar1-1 dar2-1* exhibits very small plants and leaves, but the flowers and seeds are still larger than those of the wild type (WT) [24].

UBIQUITIN-SPECIFIC PROTEASE15 (UBP15), encoded by *SUPPRESSOR2 OF DA1* (*SOD2*), regulate seed size by promoting cell proliferation in the integuments of ovules and developing seeds [25]. The *sod2*/*ubp15* mutants form small seeds, while overexpression of *UBP15* increases seed size [25]. UBP15 and DA1 have antagonistic effects in a common pathway that affects seed size, but this effect is independent of DA2 [25]. The dominant large-grain mutant, *large grain1-D* (*lg1-D*), has been shown to demonstrate a 30.8% increase in seed width and a 34.5% increase in 1000-grain weight compared to the WT [26]. Map-based cloning has shown that *LG1* encodes a constitutively expressed OsUBP15 with deubiquitinated activity in vitro [26]. Loss of function and downregulation of *OsUBP15* resulted in narrower and smaller grains than the control [26]. OsDA1 directly interacted with OsUBP15 [26]. OsUBP15 is a positive regulator of rice grain width and size. These results indicate that the ubiquitin–proteasome pathway plays an important regulatory role in seed size.

## 4. G (Guanosine Triphosphate) Protein Regulatory Pathway

G proteins are a class of signal transduction proteins that can bind with guanosine diphosphate (GDP) and possess GTPase activity [27]. The signaling pathways involving G proteins are highly conserved transmembrane signaling mechanisms in both animals and plants. When cells transduce extracellular signals, they are first received by different types of G protein-coupled receptors (GPCRs), which act as receptors for various ligands (extracellular first messengers) [27]. Subsequently, the receptors are activated, further activating heterotrimeric G proteins on the inner side of the plasma membrane. The activated G proteins then go on to activate various effectors downstream, generating second messengers within the cells. This sequential signal transduction cascade regulates the growth and development processes of organisms [27].

There are three types of Gγ proteins in rice, DEP1, GGC2, and GS3, which antagonistically regulate grain size. DEP1 and GGC2, individually or in combination, increase grain length when complexed with Gβ [28]. GS3 has no direct effect on grain size by itself, but reduces grain length by competitively interacting with Gβ [28]. The two Gγ proteins, DEP1 and GS3, antagonistically regulate grain yield. Gene editing of *DEP1* significantly increases the number of grains per panicle, while gene editing of *GS3* reduces the number of grains per panicle but significantly increases grain length [29]. Yang et al. demonstrated that *Chang Li Geng 1* (*CLG1*), encoding an E3 ligase, regulates grain size by targeting the negative regulator of grain length factor GS3 for degradation [30]. Overexpression of *CLG1* led to an increase in grain length, and CLG1 interacted with GS3 and ubiquitinated it, which then underwent degradation via the endosome degradation pathway, resulting in an increase in grain size [30]. Gγ protein AGG3, which is localized to the plasma membrane, promotes seed and organ growth by increasing the proliferation growth period in Arabidopsis [31,32]. The homologous genes of *AGG3* in rice (*GS3* and *DEP1/qPE9-1*) have been identified as important quantitative trait loci for seed size and yield [31]. GS3 and DEP1 affected seed and organ growth by inhibiting cell proliferation [31]. Overexpression of *AGG3* significantly increases the length of siliques, and the seed size and number per silique in Arabidopsis [32]. In *Camelina sativa*, constitutive or specific expression of *AGG3* in seed tissue was shown to increase seed size, seed mass, and seed number by 15–40%, effectively increasing oil yield per plant [33].

The *qLGY3* locus, which encodes a MADS-domain transcription factor OsMADS1, is associated with the quantitative and qualitative traits of rice grain yield [31]. OsMADS1 is a downstream effector of the G-protein βγ dimers and its alternatively spliced isoform OsMADS1^lgy3^ has been related to the formation of long and slender grains, potentially increasing rice yield and quality [34]. The Gγ subunits GS3 and DEP1 directly interact with the conserved keratin-like domain of MADS transcription factors and act as cofactors to enhance the transcriptional activity of OsMADS1, promoting the coordinated activation of target genes, and thereby regulating grain size and shape [34]. WIDE GRAIN 7 (WG7) activats *OsMADS1* expression by directly binding to its promoter, increasing the enrichment of histone H3K4me3 in the promoter and ultimately increasing grain size [35].

The rice genome contains a single Gα (*RGA1*) and Gβ (*RGB1*) subunit, as well as five Gγ subunits (*RGG1*, *RGG2*, *GS3*, *DEP1*/*qPE9-1*, and *GGC2*) [36]. DEP1/qPE9-1 is an atypical putative Gγ protein that controls grain size, and the density and erectness of the panicles [36]. Zhang et al. showed that rice plants carrying *DEP1/qPE9-1* have more endosperm cells per grain than those carrying *dep1*/*qpe9-1* allele [36]. Grains from the *DEP1*/*qPE9-1* line also had higher levels of ABA, auxin, and cytokinin. Exogenous application of auxin or cytokinin was shown to be able to enhance starch accumulation and the expression of grain-filling-related enzymes in *dep1*/*qpe9-1*, but had no effect on ABA [36]. *DEP1*/*qPE9-1* mainly positively regulated starch accumulation during the grain-filling stage through the auxin and cytokinin pathways, enhancing the expression of starch biosynthesis genes in the mid to late stage of grain-filling and prolonging the duration of grain filling [36].

*RGB1* is a positive regulator of cell proliferation, as evidenced by the shorter stature of *d1-5*, a mutant lacking the *RGB1* gene, in addition to the loss of *RGA1* [37]. The number of sterile seeds also increases in *RGB1* knockout lines [37]. Overexpression of *RGG2* in Nipponbare (NIP) leads to reduced plant height and smaller grain size, as *RGG2* is a negative regulator of plant growth and organ size in rice [38]. *RGG2* is also involved in GA signaling, as measured by the length of the second leaf sheath and GA-induced α-amylase activity, suggesting that *RGG2* might regulate grain and organ size through the GA pathway [38]. The upstream signals, receptors, and downstream effectors of the G-protein pathway need to be identified in future studies.

## 5. Mitogen-Activated Protein Kinase (MAPK) Pathway

MAPK cascades are conserved and involved in plant signal transduction in eukaryotes [39]. The MAPK cascades consist of three functional protein kinases: MAPKs, MAPK kinases (MAPKKs), and MAPK kinase kinases (MAPKKKs) [40]. MPK3, MPK6, and MPK10 form a small group from the MPK family in Arabidopsis [41]. The phosphorylation of MPK3/6 has been found to inactivate DA1, making it unstable for increasing the abundance of UBP15, promoting outer integument cell proliferation and increasing seed size [42]. WRKY10/MINI3 is a member of the IKU pathway related to endosperm development, and *MPK10* is specifically expressed in the early developmental endosperm, but the expression gradient is the opposite. *MPK10* and *WRKY10* inhibited each other’s expression. The inhibitory effect of *MPK10* on the expression of *WRKY10* and its downstream target genes is achieved through interaction with WRKY10, which inhibits its transcriptional activity [41]. The *mpk10* mutant produces large seeds, while WRKY10/MINI3 positively regulates seed growth [41].

Rice *OsMKK3* encodes a MAP kinase that controls grain size and chalkiness by affecting cell proliferation in the hull [43]. *OsMKKK70* was found to positively regulate rice grain size, with overexpression of *OsMKKK70* resulting in longer seeds compared with WT [44]. The *osmkkk62*/*70* and *osmkkk55*/*62/70* mutants showed significantly smaller seeds than WT, indicating functional redundancy between *OsMKKK70* and its homologs *OsMKKK62* and *OsMKKK55* [44]. OsMKKK70 was an active kinase that interacted with OsMKK4 and promoted OsMAPK6 phosphorylation [44]. Overexpression of constitutively active *OsMKK4*, *OsMAPK6*, and *OsWRKY53* partially rescued the smaller seed size of the *osmkkk62*/*70* double mutant [44]. OsMKKK70 might regulate rice grain size by activating OsMAPK6, while OsMKKK70, OsMKK4, OsMAPK6, and OsWRKY53 act together to control the signal pathway of grain shape [44].

*Dwarf and small grain 1* (*DSG1*) encodes a mitogen-activated protein kinase, OsMAPK6, which plays a critical role in rice grain size through cell proliferation, BR signaling, and homeostasis [45]. OsMAPK6 was shown to interact strongly with OsMKK4, indicating that OsMKK4 was likely to be the upstream MAPK kinase of OsMAPK6 [45]. In rice, loss of function of *OsMKKK10* has been shown to lead to small and light grains, short panicles, and semi-dwarf plants, while overexpression of the constitutively active *OsMKKK10* (*CA-OsMKKK10*) resulted in large and heavy grains, long panicles, and tall plants [46]. OsMKKK10 interacted with OsMKK4 and phosphorylated OsMKK4 [46]. OsMKK4^A227T^ encoded by the *large11-1D* allele had stronger kinase activity than OsMKK4 [46]. Plants overexpressing the constitutively active form of *OsMKK4* (*OsMKK4-DD*) also produced larger grains [46]. In summary, the data clearly indicate that the OsMKKK10-OsMKK4-OsMAPK6 signaling pathway positively regulates rice grain size and weight.

GSN1 is a negative regulator of the OsMKKK10-OsMKK4-OsMPK6 cascade that determines panicle architecture [47]. *GSN1* encodes the mitogen-activated protein kinase phosphatase OsMKP1, a dual-specificity phosphatase with unknown function [47]. Decreased *GSN1* expression was shown to lead to larger but fewer grains, while overexpression resulted in more but smaller grains [47]. GSN1 directly interacted with OsMPK6 and inactivated it by dephosphorylation [47]. The lectin receptor-like kinase LecRK-VIII.2, which acted upstream of MPK6, is a specific receptor-like kinase that regulates the final yield of Arabidopsis seeds by controlling silique number, seed size, and seed number [48]. Plants with the *lecrk-VIII.2* mutation had smaller seeds, but more siliques and seeds number, leading to increased yield [48]. However, overexpression of *LecRK-VIII.2* resulted in larger seeds but fewer siliques and seeds, resulting in a yield similar to that of WT plants [48]. These data indicate that the MAPK pathway is crucial for regulating seed size, and it is worthwhile identifying the RLKs and ligands upstream of MAPK pathway.

## 6. Transcriptional Regulators Pathway

Transcriptional regulation is crucial for multiple plant growth and developmental processes. NAC proteins belong to one of the largest families of plant-specific transcription factors (TFs), and they play important roles in plant growth and development. *TaNAC020s* are mainly expressed in developing grains [49]. Transgenic rice overexpressing *TaNAC020-B* were found to exhibit higher starch density and lower amylose contents than WT [49]. In wheat, *TaNAC020s* positively regulated starch synthesis and accumulation and were critical regulators of grain size and number [49]. Overexpression of *TaNAC100* in wheat suppresses plant height, prolongs the heading date, and promotes seed size and 1000-grain weight [50]. Transgenic rice overexpressing an *OsNAC2* mutant (*OErN*) have better plant architecture and longer panicles, and produce more grains [51]. In rice, loss of function of *OsNAC129* was found to significantly increase grain length, weight, apparent amylose content (AAC), and plant height, while overexpression of *OsNAC129* had the opposite effect [52]. The expression of *OsNAC129* was induced by ABA, and overexpression of *OsNAC129* in plants reduced sensitivity to exogenous BR, indicating that *OsNAC129* negatively regulates seed development and plant growth and participates in the BR signaling pathway.

WRINKLED1 (WRI1) is an APETALA2 (AP2) transcription factor. Two wheat *TaWRI1Ls* genes, *TaWRI1L1* and *TaWRI1L2*, were cloned based on the Arabidopsis *AtWRI1* sequence [53]. Overexpression of *TaWRI1L2* was able to compensate for the loss of *AtWRI1* function in Arabidopsis mutants, restoring seed shape and fatty acid accumulation [53]. Knockout of *TaWRI1L2* resulted in reduced grain size, 1000-grain weight, and grain fatty acid synthesis in wheat [53]. In Arabidopsis, overexpression of *RAV1*, a plant-specific B3 domain and AP2 domain-containing transcription factor, has been shown to result in reduced seed size, weight, and number per silique [54]. RAV1 repressed the expression of *MINI3* and *IKU2* by directly bonding to their promoters [54]. 

The Alfin-like (AL) family is a group of small plant-specific transcriptional factors, and null mutants of the rice *OsAl7.1* and *OsAl11* have increased seed size [55]. ATBS-INTERACTING FACTOR 2 (AIF2) is a non-DNA-binding basic helix–loop–helix (bHLH) transcription factor [56]. *AIF2ox* plants were found to have fewer siliques and fewer seeds per silique, leading to a significant decrease in total grain weight and yield [56]. Conversely, *aif2-1*/*aif4-1* plants exhibited the opposite silique numbers and seed phenotypes [56]. Positive regulation factors of seed size, such as *SHB1*, *IKU1*, and *MINI3*, were suppressed in *AIF2ox* [56]. BRASSINAZOLE RESISTANT 1 (BZR1) is an important transcription factor that regulates organ size in the BR signaling pathway in Arabidopsis [57]. *AIF2* negatively regulates BR-induced BZR1-mediated pollen development and seed formation [56]. Sucrose and BR inhibit *AIF2* ectopic accumulation, thereby increasing silique length and seed number [56]. Sucrose and BR induce *AIF2*, thus inhibiting positive regulation of pollen production and seed formation in Arabidopsis [56]. Overexpression of *ZmBZR1* in Arabidopsis results in phenotypes characterized by enlarged cotyledons, rosette leaves, floral organs, and seed size [57]. The cells in transgenic *ZmBZR1* lines are significantly larger in the rosette leaves and other organs than in those of the WT [57]. *ZmBES1/BZR1-5* was found to positively regulate grain size, as overexpression of *ZmBES1/BZR1-5* significantly increased seed size and weight in Arabidopsis and rice [58]. ZmBES1/BZR1-5 bound to the promoters of *AP2/EREBP*, inhibiting transcription [58]. ZmBES1/BZR1-5 interacted with casein kinase II subunit β4 (ZmCKIIβ4) and ferredoxin 2 (ZmFdx2) in vitro and in vivo [58].

GATA represents a highly conserved family of transcription factors, and *OsGATA8* has been shown to increase seed size and tolerance to abiotic stresses in Arabidopsis and rice, as well as helping to maintain yield under stress, with overexpressing plants having a yield about 46% higher than the WT [59]. *WRKY6*, a WRKY6 family transcription factor, has a high expression level in developing seeds and was shown to play an important role in regulating the accumulation of fatty acids (FAs) in Arabidopsis [60]. Mutation of *WRKY6* led to significantly increased seed size, accompanied by increased FA content and changes in FA composition [60]. In Arabidopsis, MYB56, an R2R3 MYB transcription factor, was shown to positively control seed size, since loss-of-function or knock-down of *MYB56* yielded smaller seeds, while overexpression of *MYB56* produced larger seeds than the WT [61]. Microscopic observation showed that *myb56* resulted in the formation of smaller endothelial cells with reduced numbers of cells in the outer integument, whereas *MYB56* overexpressing lines had expanded endothelial cells and increased numbers of cells in the outer integument [61]. MYB37, an R2R3 MYB subgroup 14 transcription factor, has been shown to play a positive role in the regulation of seed production [62]. Overexpression of *MYB37* delayed the flowering time, as the appearances of flower buds in the OE1 and OE6 transgenic plants were delayed 8 and 12 days, respectively, under long day conditions [62]. However, mature OE1 and OE6 plants exhibited greater growth than the WT plants in the later developmental stages, with significantly higher stem height and weight or bigger biomass. Furthermore, transgenic OE1 and OE6 plants produced more siliques, leading to higher seed production than the WT plants [62].

Growth-regulating factor (GRF) interacting factor 1 (GIF1) is a plant-specific transcriptional cofactor that positively regulates the grain size in rice [63]. Overexpression and functional knock-out using CRISPR/Cas9 strategies showed that *OsGIF1* not only positively regulates the size of rice leaves, stems, and grains, but also affects rice reproduction [63]. *Grain length and width 2* (*GLW2*) encodes growth-regulating factor 4 (OsGRF4), which was shown to be regulated by miR396c in vivo [64]. Mutation in *OsGRF4* disrupted the targeted regulation of *OsGRF4* by OsmiR396, resulting in larger seeds and increased grain yield [64]. OsGIF1 directly interacted with OsGRF4, increasing its expression to improve seed size [64]. These data indicate that transcriptional regulation is important for regulating seed size.

## 7. Phytohormone Regulatory Pathways

### 7.1. Auxin Regulatory Pathway

Auxin is an essential phytohormone in plant development [65] that is mainly involved in regulating different signaling pathways in embryos, endosperm, and seed coat, leading to various changes in seed size or weight [8].

Rice YUCCA (YUC) flavin-containing monooxygenase encoding gene *OsYUC11* has been shown to be a key factor for auxin biosynthesis in endosperm [65]. Both *osyuc11* and *osnf-yb1* showed reduced seed size and increased chalkiness, accompanied by reduced indole-3-acetic acid level [65]. OsYUC11-mediated auxin biosynthesis was found to be critical for endosperm development in rice [65]. Mutations of the wheat Aux/IAA gene *TaIAA21* were found to significantly increase grain length, width, and weight [66]. *Auxin Response Factor 12* of maize (*ZmARF12*), which mediates the expression of auxin-responsive genes, was found to regulate seed size development, with the mutant of *ZmARF12* exhibiting a larger seed size and a higher grain weight [67]. Mutations of the tetraploid wheat *ARF25* gene were shown to significantly reduce grain size and weight [66]. Transcriptional activation experiments showed that *ARF25* promoted the transcription of *ERF3*, while *TtERF3* mutations led to reduced grain size and weight [66].

The modified expression of *TaCYP78A5* can accumulate auxin, increasing the wheat grain weight and yield per plant [68]. Constitutive overexpression of *TaCYP78A5* has been found to lead to significantly increased seed size and weight, but not yield, per plant due to the enhancement of apical dominance [68]. *JcARF19* has been demonstrated to control seed size and seed yield in the woody plant *Jatropha curcas* [69]. Overexpression of *JcARF19* significantly increased seed size and seed yield in Arabidopsis and *Jatropha curcas*, indicating the importance of the auxin pathway in controlling seed yield [69]. These data indicate that the synthesis, transport, and response of auxin are important for regulating seed size.

### 7.2. Brassinosteroid (BR) Regulatory Pathway

BRs are a group of plant steroid hormones that play an important role in regulating organ size [70]. Sun et al. identified an ortholog of rice *DWARF11* (*D11*) in maize called *ZmD11* [70]. They found that constitutive expression of *ZmD11* significantly increased seed length, seed area, seed weight, and both seed starch and protein contents in rice and maize [70]. Overexpression of the BR-synthesis gene *D11* or the BR signaling factor *OsBZR1* has been found to lead to the accumulation of more sugar in developing anthers and seeds, resulting in higher grain yield [71]. Conversely, knockdown of *D11* or *OsBZR1* led to defective pollen maturation, reduced seed size and weight, and less starch accumulation [71]. The *CLUSTERED PRIMARY BRANCH 1* (*CPB1*) gene is a new allele of *D11* in rice. Mutations in the *CPB1*/*D11* gene specifically affect rice panicle architecture and seed size development, and *CPB1*/*D11* transgenic plants driven by panicle-specific promoters can possess enlarged seed size and enhanced grain yield [72]. *TaD11* has been shown to affect wheat grain size and root length [73]. Ectopic expression of *TaD11-2A* rescued the abnormal panicle structure and plant height of the *cpb1* mutant, and also increased endogenous BR levels, resulting in improved grain yield and quality in rice [73]. The *tad11-2a-1* mutant showed dwarfism, smaller grains, and reduced endogenous BR levels [73]. The semi-dominant mutant *Small and round seed 5* (*Srs5*) encodes an alpha-tubulin protein [74]. The mutant in the BR receptor BRI1 (*d61-2*) results in a short seed phenotype due to impaired cell length [74]. The seeds of the *Srs5 d61-2* double mutant are smaller than those of either single mutant [74]. Overexpression of *SRS5* promotes grain length, and can rescue the shortened grain length of BR-related mutants [75].

BR plays a crucial role in determining the seed size, quality, and shape by transcriptionally modulating specific seed developmental pathways. The seeds of BR-deficient mutant *de-etiolated 2* (*det2*) were shown to exhibit decreased seed cavity, reduced endosperm volume and integument cell length, resulting in smaller and less elongated seeds compared with WT [76]. The *det2* mutant also showed delayed embryo development, with reduced embryo cell size and number [76]. BR activated the expression of positive regulators of seed size, such as *SHB1*, *MINI3* and *IKU2*, while suppressing the expression of negative regulators of seed size, such as *AP2* and *ARF2* [76]. In rice, glycogen synthase kinase-2 (GSK2) was found to phosphorylate WRKY53 and reduce its stability, indicating that WRKY53 is a substrate of GSK2 in BR signaling [77]. WRKY53 interacted with BZR1 and cooperated to regulate BR-related developmental processes [77]. WRKY53 and the MAPKKK10-MAPKK4-MAPK6 cascade pathway jointly participated in controlling leaf angle and seed size, suggesting that WRKY53 was a direct substrate of this pathway [77]. GSK2 phosphorylated MAPKK4 to inhibit MAPK6 activity, suggesting that GSK2-mediated BR signaling might also regulate the MAPK pathway [77]. These data suggest that BRs promote seed growth.

### 7.3. Gibberellin (GA) Regulatory Pathway

GAs are important phytohormones that play a crucial role in controlling seed germination, stem elongation, leaf development, and floral induction [78]. Transgenic plants overexpressing Arabidopsis GA2-oxidase gene (*AtGA2ox8*) in *Brassica napus* L. were shown to exhibit a significant increase in seed yield by 9.6–12.4% [79]. Gibberellic acid-stimulated Arabidopsis 4 (GASA4) is one of the 14 members of the small polypeptide family in Arabidopsis, regulating flowering and seed development and affecting seed size, weight, and yield [80]. The *gasa4-1* null mutant significantly reduces seed weight, while the *35S::GASA4*-overexpressing lines significantly increase seed weight and yield [80]. Even in *gasa4-1* plants with smaller seeds, higher total seed yields have also been observed compared to WT due to increased branching, resulting in a greater seed number [80]. *TaGW2-6A* encodes RING E3 ubiquitin-ligases, which are involved in GA signal transduction, and negatively regulate grain size [81]. *TaGW2-6A* allelic variation increases seed size through the GA signaling regulatory pathway [81]. *TaGW2-6A* allelic variation regulates GA synthesis through GA3-oxidase, resulting in high expression of *GASA4* to control endosperm cell elongation and division during grain filling [81].

### 7.4. Jasmonate (JA) Regulatory Pathway

JA is a key phytohormone essential for multiple developmental processes in plants [82]. Seed size is regulated positively by the JA pathway factors COI1, MYC2 (and its homologs), MED25, and JAR1, but promoted by the JA signaling repressor JAZ [82,83]. It has been found that the JA receptor mutants, both *coi1-2* and *coi1-8*, have obviously enlarged seeds compared to WT in Arabidopsis, and the OsCOI1-RNAi lines *coi1-13* and *coi1-18* also exhibited much larger grains than WT in rice [82]. MYCs are key transcription factors responsible for diverse JA responses [82]. The *myc2* single mutant, *myc2*/*3* double mutant, *myc2*/*3*/*4* triple mutant, and *myc2*/*3*/*4*/*5* quadruple mutant all exhibited a large seed size phenotype, with increased 100-grain weight, seed length and width compared with WT, indicating that *MYC2*, *MYC3*, *MYC4*, and *MYC5* acts redundantly to repress seed size [82]. The MYC-MED25/PFT1 complex was required for the activation of JA-responsive gene expression. The *med25* mutants, both *pft1-2* and *pft1-3*, displayed a significant increase in seed size and 100-grain weight [82]. 

JA has been shown to stimulate the degradation of JASMONATE ZIM-DOMAIN (JAZ) protein, thereby relieving the inhibition of transcription factors coincident with reduced growth and fecundity. The mutant of JA signaling repressor *JAZ6* (*jaz6-3*) exhibited decreased seed length, width and 100-grain weight compared to WT [82]. The JA signaling repressor *OsJAZ11* has been shown to be involved in regulating seed width and weight, since transgenic plants overexpressing *OsJAZ11* showed up to a 14% increase in seed width and a ~30% increase in seed weight compared to WT [83]. It was demonstrated that OsJAZ11 interacted with OsMADS29 and OsMADS68. *OsGW7*, a key negative regulator of grain size, was downregulated in *OsJAZ11*-overexpressing plants, suggesting that *OsJAZ11* participated in regulating seed size by coordinating the expression of JA-related, *OsGW7*, and *MADS* genes [83]. In Arabidopsis, JAZ-deficient mutants (*jazD*) with mutations in three *MYC* genes (*mycT*) were found to exhibit enhanced defense and reduced seed yield [84]. The de-repression of *MYCs* in *jazD* resulted in reduced fruit size and seed yield [84]. JA plays an essential role in plant seed development, but the regulation of seed size by JA is still not very clear.

### 7.5. Cytokinin (CK) Regulatory Pathway

CKs are a group of phytohormones that play important roles in multiple biological processes [85] and are crucial for determining grain yield [86]. Exogenous application of CKs, ectopic expression of IPT, or downregulation of cytokinin oxidase/dehydrogenase (CKX) lead to increased seed yield [87]. ONAC096 regulates grain yield by affecting leaf senescence and panicle number in rice [88]. Inhibition of *OsCKX2* expression in *onac096* mutants was reported to lead to a 15% increase in panicle number [88]. *OsCKX2* regulates rice grain yield by controlling cytokinin levels and modulating floral primordial activity under normal and abiotic stress conditions [86]. Yeh et al. used artificial short hairpin RNAs (shRNA) to specifically silence *OsCKX2*, and found that both CX3 and CX5 transgenic plants produced more tillers and grains per plant and had a heavier 1000-grain weight than WT [89]. In cotton, suppression of *CKX* was found to lead to increased endogenous CKs levels and the transgenic plants with moderate inhibition resulted in delayed leaf senescence, enhanced photosynthesis, increased fruiting branches and bolls, and larger seed size [90]. 

Treatment with thidiazuron (TDZ), a novel and efficient cytokinin, was shown to increase seed diameter and silique length, as well as the yield, since the embryo and cotyledon epidermal cells were obviously larger in TDZ-treated *Brassica napus* plants [91]. TDZ upregulates genes related to maternal tissue development, including two G-protein signaling genes (*AGG3* and *RGA1*) and two transcription factors (*ANT* and *GS2*) and might regulate key genes involved in auxin metabolism and endosperm growth, resulting in enlargement of cotyledon epidermal cells and seed size in *Brassica napus* [91].

Arabidopsis histidine kinases (AHKs) are the receptors in the CK signaling pathway. *ahk2 ahk3* mutants showed an increase in seed size but a decrease in seed yield, indicating that CKs negatively regulate seed size and positively regulate seed yield [92]. Bartrina et al. isolated the constitutively active gain-of-function variants of *AHK2* and *AHK3* genes, named repressor of cytokinin deficiency2 (*rock2*) and *rock3* [93]. The transgenic *AHK2: rock2* lines showed almost twice as many siliques as the WT. The number and yield of siliques were increased in *AHK3: rock3* due to an extended flowering period [93]. Riefler et al. found that the seed length and width of the *ahk2 ahk3 ahk4* triple mutant were 30% larger than those of WT, while the seed volume of *ahk2 ahk3 ahk4* increased to ~250% of the WT seed volume [94].

*PURINE PERMEASE1* (*OsPUP1*) is involved in importing CKs into the endoplasmic reticulum (ER) to unload CKs from the vascular tissues through cell-to-cell transport, and its overexpression results in reduced leaf length, plant height, grain weight, panicle length, and grain number in rice [85]. A dominant mutant, *big grain3* (*bg3-D*), shows a significant increase in grain size as a result of activation of *OsPUP4*, which is involved in the long-distance transport of CKs [95]. The T-DNA mutant of *OsPUP7*, which is involved in transporting the CK derivative caffeine, exhibits various phenotypic changes, such as increased plant height, larger seeds, and delayed flowering [96]. These results indicate that the PUP transport system, which is involved in CK transport, is important for seed development. Taken together, these data clearly indicate that CKs are negative regulators of seed size.

### 7.6. Abscisic Acid (ABA) Regulatory Pathway

ABA regulates various aspects of plant growth and development as well as responses to abiotic stress [97]. *LOS5*/*ABA3* is involved in ABA biosynthesis by encoding molybdenum co-factor sulfurase, which is required by aldehyde oxidase (AO) in the last step of ABA biosynthesis in plants [98]. Transgenic plants overexpressing *LOS5*/*ABA3* have been reported to show at least a 21% increase in seed yield compared to the WT under drought stress conditions [98]. *OsAO3* is essential for regulation of grain yield in rice, since *osao3* mutant increases grain yield, while overexpression of *OsAO3* reduces grain yield by affecting panicle number per plant, spikelet number per panicle, and spikelet fertility [97]. Cytosolic ABA receptors PYRABACTIN RESISTANCE 1 LIKE/REGULATORY COMPONENTS OF ABA RECEPTORS (PYL/RCARs) can regulate ABA-dependent gene expression in rice [99]. Constitutive expression of *OsPYL/RCAR5* slightly reduces plant height and severely decreases seed yield under paddy field conditions, although abiotic stress tolerance is improved [99]. 

ABSCISIC ACID DEFICIENT2, a unique short-chain dehydrogenase/reductase involved in ABA biosynthesis, has been reported to regulate seed development, since the *aba2-1* mutant exhibited increased seed size, mass, and embryo cell number, but endosperm cellularization was delayed [100]. The RNA levels of *SHB1*, *MINI3*, and *IKU2*, which are involved in seed size control, were significantly increased in *aba2-1*, and *ABSCISIC ACID-INSENSITIVE 5* (*ABI5*), which is involved in ABA signaling, was decreased in *aba2-1* [100]. Functional loss of *ABI5* led to increased seed size. *ABA2* regulated the endogenous levels of ABA and controlled the transcription of *ABI5*. *ABI5* inhibited the expression of *SHB1*, which further regulated *MINI3* and *IKU2* [100]. These data indicate that the synthesis and response of ABA are important for regulating seed size.

## 8. MicroRNAs (miRNAs) Regulatory Pathway

miRNAs are a class of small noncoding RNAs that play key regulatory roles in seed development. For example, overexpression of maize zma-miR169o increases seed size and weight, whereas inhibiting its expression has the opposite effect [101]. Zma-miR169 negatively regulates its target gene *ZmNF-YA13*, which plays a crucial role in determining seed size and regulates the expression of the auxin biosynthesis gene *ZmYUC1*, thereby regulating maize seed size and, ultimately, yield [101]. OsmiR530 negatively regulates grain yield, as suppressing OsmiR530 increases grain yield, while overexpression of OsmiR530 significantly reduces grain size and panicle branching, leading to a decrease in yield [102]. *Phytochrome-interacting factor-like 3* (*OsPL3*) encodes a protein containing a PLUS3 domain, and is directly targeted by OsmiR530, while knocking out *OsPL3* reduces grain yield [102]. Transgenic plants both overexpressing (miR529a-OE) and suppressing (miR529a-MIMIC) miR529a result in narrower grains and a lower grain weight, while miR529a-OE plants produce longer and more slender grains, whereas miR529a-MIMIC plants produces shorter grains [103]. miR529a regulates grain size by altering the expression of *SQUAMOSA PROMOTER BINDING-LIKE* (*SPL*) transcription factors, which can control rice tiller, panicle architecture and grain size [103]. 

By using Short Tandem Target Mimic (STTM), Zhao et al. successfully suppressed the expression of mature miR159 in STTM159 transgenic plants, leading to increased expressions of its two target genes, *OsGAMYB* and *OsGAMYBL1* (*GAMYB-LIKE 1*) [104]. Moreover, STTM159 plants were dwarfed, with reduced organ size, stem diameter, length of flag leaf, main panicle, spikelet hulls and grain size [104]. Transgenic plants overexpressing miR398 (OX-miR398a) have been shown to increase rice panicle length, grain number and grain size; however, silencing of miR398 (STTM398) had the opposite effects [105]. *Os07g46990*, one of the miR398 targets, was modified, and the mOs07g46990 transgenic plants showed decreased panicle length and grain number per panicle like the phenotypes of STTM lines [105]. Wang et al. knocked down miR160 and miR165/166 (STTM160 and STTM165/166), and found that STTM160 showed significantly smaller grain size, lower grain weight, shorter silique length, shorter plant height, and more serrated leaves, while STTM165/166 showed reduced seed numbers, disabled siliques, and upward-curled leaves [106]. These data indicate that regulation of miRNAs is essential for seed development.

## 9. Conclusions and Prospects

Seed size is particularly important for agricultural production and is regulated by the coordinated control of the ovule, endosperm, and embryo, while the development of these tissues is regulated by various pathways (Figure 1). The ubiquitin–proteasome pathway and the MAPK signaling pathway primarily influence integument development. The IKU pathway mainly regulates endosperm development. Furthermore, cross-talk may occur among several pathways involved in seed size regulation through certain genes or phytohormones, such as the auxin signaling pathway and the BR signaling pathway (*ARF2* and *BZR1*), the IKU pathway and the cytokinin signaling pathway (*MINI3* and *CKX2*), the BR signaling pathway and the MAPK pathway (*OsMKK4*, *OsMAPK6*, and BR-related genes), and the G-protein pathway and the BR signaling pathway (Gα influences BR signal cascade). However, despite this, the molecular mechanisms governing seed size regulation are still not fully understood, and the connections between various pathways are scarce, thus failing to elucidate how the development of the ovule, endosperm, and embryo are coordinated with each other.

Seed size is regulated by a complex regulatory network, and although many genes have been identified (Table 1), the regulatory mechanisms are still not well understood, and there is still much to be explored regarding upstream and downstream genes and proteins. For instance, the regulatory processes of some genes are still unclear, and their target substrates remain unknown. Are they involved in signal transduction processes? What are the specific ligands and regulatory factors? Do these regulatory genes exist in other crops? How are they expressed during seed development? Modern biotechnologies such as gene editing techniques and systems biology approaches like transcriptomics, proteomics, metabolomics, and epigenomics will facilitate researchers in unraveling the molecular mechanisms that regulate seed size more rapidly.

Seeds are the most crucial source of food for human survival. The study of genes regulating seed size development is of significant theoretical value, and provides essential genetic information for improving food production through genetic engineering techniques. It holds great potential for various areas, such as functional genetic improvement, and offers important prospects for applications in increasing crop yield and other related fields.

## Figures and Tables

**Figure 1 ijms-24-10666-f001:**
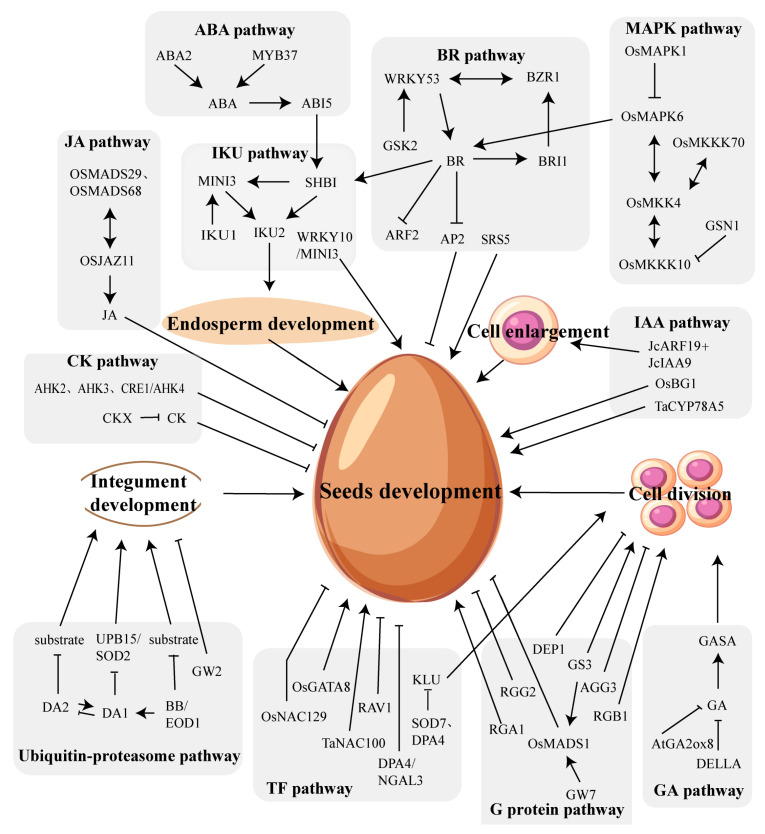
Various signaling pathways involved in seeds development.

**Table 1 ijms-24-10666-t001:** Regulatory factors involved in seeds development.

Pathway	Protein	Species	Phenotype of Loss of Function	Phenotype of Gain of Function	Reference
IKU pathway	IKU1	*Arabidopsis thaliana*	Small seeds	-	[11]
	IKU2	*Arabidopsis thaliana*	-	Big seeds	[12]
	MINI3	*Arabidopsis thaliana*	Small seeds	-	[13]
	SHB1	*Arabidopsis thaliana*	Small seeds	Big seeds	[14]
Ubiquitin–proteasome pathway	GhGW2-2D	*Gossypium* spp.	-	Small seeds	[17]
	AtDA1	*Arabidopsis thaliana*	Big seeds	-	[19]
	BnDA1	*Brassica napus*	Big seeds	-	[19]
	TaDA1	*Triticum aestivum*	Big seeds	Small seeds	[20]
	SOD7	*Arabidopsis thaliana*	-	Small seeds	[21]
	GhDA1-1A	*Gossypium* spp.	-	Big seeds	[22]
	DA2	*Arabidopsis thaliana*	Big seeds	Small seeds	[23]
	UBP15	*Arabidopsis thaliana*	Small seeds	Big seeds	[25]
	OsUBP15	*Oryza sativa*	Small seeds	-	[26]
	GW2	*Oryza sativa*	-	Small seeds	[26]
G protein regulatory pathway	DEP1	*Oryza sativa*	-	Big seeds	[28]
	GGC2	*Oryza sativa*	-	Big seeds	[28]
	GS3	*Oryza sativa*	Big seeds	-	[28]
	CGL1	*Oryza sativa*	-	Big seeds	[30]
	AGG3	*Arabidopsis thaliana*	Big seeds	-	[32]
	RGG2	*Oryza sativa*	-	Small seeds	[38]
MAPK pathway	MPK10	*Arabidopsis thaliana*	Big seeds	-	[41]
	OsMKK3	*Oryza sativa*	-	Longer seeds	[43]
	OsMKKK70	*Oryza sativa*	-	Longer seeds	[44]
	OsMKKK10	*Oryza sativa*	Small seeds	Big seeds	[46]
	OsMKK4	*Oryza sativa*	-	Big seeds	[46]
	OsMKP1	*Oryza sativa*	Big seeds	Small seeds	[47]
Transcriptional regulators pathway	TaNAC100	*Triticum aestivum*	-	Big seeds	[50]
	OsNAC129	*Oryza sativa*	Big seeds	Small seeds	[52]
	TaWRI1L2	*Triticum aestivum*	Small seeds	-	[53]
	RAV1	*Arabidopsis thaliana*	-	Small seeds	[54]
	OsAL7.1	*Oryza sativa*	Big seeds	-	[55]
	OsAL11	*Oryza sativa*	Big seeds	-	[55]
	AIF2	*Arabidopsis thaliana*	-	Fewer seeds	[56]
	ZmBZR1	*Arabidopsis thaliana*	-	Big seeds	[57]
	ZmBES1/BZR1-5	*Arabidopsis thaliana*	-	Big seeds	[58]
	OsGATA8	*Oryza sativa*	-	Big seeds	[59]
	WRKY6	*Arabidopsis thaliana*	Big seeds	-	[60]
	MYB56	*Arabidopsis thaliana*	Small seeds	Big seeds	[61]
	MYB37	*Arabidopsis thaliana*	-	Higher seed yield	[62]
	GIF1	*Oryza sativa*	Small seeds	Big seeds	[63]
	OsGRF4	*Oryza sativa*	Big seeds	-	[64]
Auxin regulatory pathway	OSYUC11	*Oryza sativa*	Small seeds	-	[65]
	TaIAA21	*Triticum aestivum*	Big seeds	-	[66]
	TtARF25	*Triticum turgidum*	Small seeds	-	[66]
	ZmARF12	*Zea mays*	Big seeds	-	[67]
	TaCYP78A5	*Triticum aestivum*	-	Big seeds	[68]
	JcARF19	*Jatropha curcas*	-	Big seeds	[69]
BRs regulatory pathway	ZmD11	*Zea mays*	-	Big seeds	[70]
	D11	*Oryza sativa*	Small seeds	Higher grain yield	[71]
	OsBZR1	*Oryza sativa*	Small seeds	Higher grain yield	[71]
	CPB1	*Oryza sativa*	-	Big seeds	[72]
	TaD11-2A	*Triticum aestivum*	Small seeds	Big seeds	[73]
	D61-2	*Oryza sativa*	Short seeds	-	[74]
	SRS5	*Oryza sativa*	-	Big seeds	[75]
	DET2	*Arabidopsis thaliana*	Small seeds	-	[76]
	WRKY53	*Oryza sativa*	-	Big seeds	[77]
GA regulatory pathway	GA2ox8	*Arabidopsis thaliana*	-	Higher seed yield	[79]
	GASA4	*Arabidopsis thaliana*	Small seeds	Big seeds	[80]
JA regulatory pathway	COI1	*Arabidopsis thaliana*	Big seeds	-	[82]
	MYC2	*Arabidopsis thaliana*	Big seeds	-	[82]
	MED25	*Arabidopsis thaliana*	Big seeds	-	[82]
	JAZ6	*Arabidopsis thaliana*	Small seeds	-	[82]
	OsJAZ11	*Oryza sativa*	-	Big seeds	[83]
CK regulatory pathway	OsCKX2	*Oryza sativa*	More grains	-	[89]
	AHK2/3/4	*Arabidopsis thaliana*	Big seeds	Higher seed yield	[92,93,94]
	OsPUP1	*Oryza sativa*	-	Reduced grain weight and number	[85]
	OsPUP4	*Oryza sativa*	-	Big seeds	[95]
	OsPUP7	*Oryza sativa*	Big seeds	-	[96]
ABA regulatory pathway	LOS5/ABA3	*Arabidopsis thaliana*	-	Higher seed yield	[98]
	OsAO3	*Oryza sativa*	Higher yield	Reduced grain yield	[97]
	OsPYL/RCAR5	*Oryza sativa*	-	Reduced grain yield	[99]
	ABA2	*Arabidopsis thaliana*	Big seeds	-	[100]
	ABI5	*Arabidopsis thaliana*	Big seeds	-	[100]
miRNA regulatory pathway	zma-miR169o	*Zea mays*	Small seeds	Big seeds	[101]
	OsmiR530	*Oryza sativa*	Higher yield	Small seeds	[102]
	miR529a	*Oryza sativa*	Narrower grain	Narrower and longer grain	[103]
	miR159	*Oryza sativa*	Small seeds	-	[104]
	miR398	*Oryza sativa*	Small seeds	Big seeds	[105]
	miR160	*Triticum aestivum*	Small seeds	-	[106]
	miR165/166	*Triticum aestivum*	Reduced seed number	-	[106]

## Data Availability

Not applicable.

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
