# Peer review of "Molecular Network for Regulation of Seed Size in Plants"

_ijms, 2023, doi:10.3390/ijms241310666_

Round 1
Reviewer 1 Report
The control of seed size is a major concern in crop breeding and has been the subject of enormous human effort since ancient times. In addition to traditional breeding research, the mechanisms of seed size regulation are being revealed in detail through forward and reverse genetics using model plants and through the progress of genome sequencing.
This paper summarises the network of seed size regulation, including the latest studies, and suggests strategies to improve yields. The content of the paper is evident and is organized in a way that makes it easy for researchers in the related fields to refer to it. No particular changes were found to be required.
Author Response
Thanks very much for your careful review.
Reviewer 2 Report
In general, the paper is well written (although some parts need extra proofreading and editing, several examples are indicated below). The review is quite exhaustive and complete.
As a criticism, I miss some conclusions in each section of the review, so the non-experts, those not interested in each particular gene, can have take-home messages. The review is a quite complete and exhaustive recompilation of references, but it is hard to follow, and it lacks some general summary for each section. For example, is this pathway or this hormone promoting increase in seed size or is it detrimental? For example, something like “All together, data clearly indicates that CKs are positive regulators of seed size”. Just make it easier for the reader.
Other issues to correct:
-In line 22-23, modify or clarify “seed shape also plays an important role in seed development and weight [2]”. Although in the abstract of reference #2 authors say that “seed shape also contributes to seed development and weight”, I believe that this idea is nor correct. I should recommend changing to “seed development also plays an important role in seed shape and weight” and remove the reference. To me, seed shape, size, and weight is a consequence of seed development.
-I would recommend to re-elaborate the ideas in lines 31- 35: “Transcription factors regulate cell growth in the ovule, resulting in seed growth. Plant hormones not only regulate plant growth and development, but also regulate seed size [10]. In rice, grain size is the main determinant of yield, and the spikelet hull is a key determinant of grain size [11].” For example. Move these sentences later in the paragraph, after mentioning the miRNAs, and modify to something like: “ In addition to these well-known pathways, transcription factors and plant hormones have been involved in the control of ovule and seed size. Moreover, the correct development of other plant structures is also key determinant of grain size and hence of crop yield, such as the spikelet hull in rice”. This is just an idea on how to improve this paragraph.
-I would recommend one-two sentences at the beginning of the sections 2, 3, 4, and 5, explaining what these pathways are and their function at the cellular-biochemical level as well as during plant development, not only the role of individual genes in seed development. They have done a brief introduction to the MAP kinase pathway (section 5).
-Move sentence in line 183-184 (The OsMKKK10-OsMKK4-OsMAPK6 signaling pathway positively regulated rice grain size and weight) at the beginning of the paragraph. In this way, they first introduce the conclusion or message and elaborate it afterwards. If not, change the sentence to something like “In summary, data clearly indicate that OsMKKK10-OsMKK4-OsMAPK6 signaling pathway positively regulated rice grain size and weight”.
-Rewrite sentence in line 221-222 to “The Alfin-like (AL) family was a group of small plant-specific transcriptional factors. Rice mutants al7.1 and al11 have increased seed size” or “The Alfin-like (AL) family was a group of small plant-specific transcriptional factors, and null mutants of the rice OsAL7.1 and OsAL11 have increased seed size”.
-Revise the text to check for mutant/gene names. For example, in lane 263, “ZmARF12 mutant” is not correct; either use “zmarf12 mutant”, “maize arf12 mutant” or “mutant of ZmARF12”.
-What is the relationship among ARF12, GIF1, and GRF in paragraph lines 261-271? Unless it is justified, I would move the data for ARF12 to the auxin section 7.1, where they mention data for other ARFs, like wheat ARF25 and JcARF19.
-Add maize in the list of plants in lines 40-41, and remove the plant name (Zea mays) in line 295-296. Change sentence to “This article reviews the research results on the mechanisms that regulate seed size in plants such as Arabidopsis (Arabidopsis thaliana L.), wheat (Triticum aestivum L.), rice (Oryza sativa L.), maize (Zea mays L.), and other plants.
-Please, revise the sentence in lines 340-342 “TaGW2-6A encoded a RING E3 ubiquitin-ligases with actively involved in GA perception and signal transduction and negatively regulated grain size [79].” First, the word “with” should be eliminated, but moreover it is not clear to me that a E3 ubiquitin-ligase is involved in GA perception. The author of reference #79 mentioned this in the abstract, but they do not refer to TaGW2-6A in particular. I would change to “TaGW2-6A encodes a RING E3 ubiquitin-ligases, which are involved in GA signal transduction, and negatively regulated grain size [79].
-Line 375. CK is not a singular molecule (line 375); cytokinins (CKs) are a group of phytohormones. Change accordingly. Same for GAs (line 331).
-Line 461, what is the nature of the mOs07g46990 transgenic plants? Is it a knockout/knockdown mutant, or and overexpressor?
-Line 477, remove “and so on” and elaborate the idea. It is too informal.
-I would also recommend introducing Figure 1 in all sections (from section 2), as it may help to follow the text, not only at the end. It is the only figure and is only mentioned briefly at the end of the text.
-Figure 1, unless better explained, GA2ox8 would reduce GA levels, not affect directly seed size independently of the GAs. In addition, modify to GA2ox8 (not GA20x8), as it may induce some confusion to the GA20 oxidases. And very important, GAs negatively regulate DELLA protein stability, not as indicated in the figure. I recommend revising very carefully all Figure 1 to look for similar mistakes in other pathways.
Other text issues
-It seems that all italics of the text have been eliminated. Change to italics all plant names, gene names, and mutant names.
-Some corrections must also be done in the text. I have some examples. I recommend a complete revision and editing of the text.
For example:
-Line 46 Change to “IKU1 encodes a protein containing the plant-specific VQ motif”.
-Line 261, rewrite “a member transcription factor”. Something is missing in this sentence.
-Line 289, ARFs have already been spelled out before (in line 261). No need to do it again here.
-Line 331, change to “GAs are important phytohormones that play a crucial role...”
-Line 350, missing comma after coi1-8? Also missing comma after triple mutant in line 354, and after MYC4 in line 355. Again, add comma after pft1-3 in line 358. Revise all text.
-Line 388-391; Rephrase to “Treatment of thidiazuron (TDZ), a novel and efficient cytokinin, increased seed diameter and silique length, as well as the yield, since the embryo and cotyledon epidermal cells were obviously larger in TDZ-treated Brassica napus plants [89].
-Line 395, change “Arabidopsis histidine kinases (AHKs) are receptors in the CK signaling pathway” to “Arabidopsis histidine kinases (AHKs) are CK receptors”, or “Arabidopsis histidine kinases (AHKs) are the receptors in the CK signaling pathway”.
-Line 413, change critical to important.
-Line 437, sentence should begin with lower case “miRNAs are a class…”. Same in line 451 for miR529a, and through the text.
-Line 437-239, rewrite “MiRNAs are a class of small noncoding RNAs that play key regulatory roles in plant seed development, and the Zma-miRNA169 family is highly expressed during maize seed development [99]. Overexpression of maize zma-miR169o increased seed size and weight, while inhibiting its expression had the opposite effect [99]”. Change to something like: “miRNAs are a class of small noncoding RNAs that play key regulatory roles in seed development. For example, overexpression of maize zma-miR169o increased seed size and weight, whereas inhibiting its expression had the opposite effect [99]”.
-Line 448-450, this sentence is quite confusing; rewrite. “both transgenic plants overexpressing (miR529a-OE) and suppressing (miR529a-MIMIC) the miR529a resulted in narrower grain and lower grain weight, miR529a-OE plants produced longer and slenderer grains, whereas miR529a-MIMIC plants produced shorter grains [101]”.
-Line 454-458, rewrite to indicate what exactly STTM159 is: “By using Short Tandem Target Mimic (STTM), Zhao et al. successfully suppressed the expression of mature miR159 in STTM159 transgenic plants, leading to increased expressions of its two target genes, OsGAMYB and OsGAMYBL1 (GAMYB-LIKE 1) [102]. Moreover, STTM159 plants were dwarfed with reduced organ size, stem diameter, length of flag leaf, main panicle, spikelet hulls and grain size [102].
These are only few examples, I did not go through all the text, so I recommend a deep proofreading/editing of the text.
Some corrections must also be done in the text. I have included some issues/suggestions in my review, but strongly recommend a complete revision and editing of the text.
Author Response
Thanks very much for the careful review.
Response to Reviewer 2 Comments
In general, the paper is well written (although some parts need extra proofreading and editing, several examples are indicated below). The review is quite exhaustive and complete.
As a criticism, I miss some conclusions in each section of the review, so the non-experts, those not interested in each particular gene, can have take-home messages. The review is a quite complete and exhaustive recompilation of references, but it is hard to follow, and it lacks some general summary for each section. For example, is this pathway or this hormone promoting increase in seed size or is it detrimental? For example, something like “All together, data clearly indicates that CKs are positive regulators of seed size”. Just make it easier for the reader.
Response 1: Thanks very much for your careful review. We added general summary for each section in the revised manuscript (marked in red).
Other issues to correct:
-In line 22-23, modify or clarify “seed shape also plays an important role in seed development and weight [2]”. Although in the abstract of reference #2 authors say that “seed shape also contributes to seed development and weight”, I believe that this idea is nor correct. I should recommend changing to “seed development also plays an important role in seed shape and weight” and remove the reference. To me, seed shape, size, and weight is a consequence of seed development.
Response 2: Thanks for your advice. We had modified “seed shape also plays an important role in seed development and weight [2]” to “seed development also plays an important role in seed shape and weight” based on your suggenstion, and the reference [2] was removed.
-I would recommend to re-elaborate the ideas in lines 31- 35: “Transcription factors regulate cell growth in the ovule, resulting in seed growth. Plant hormones not only regulate plant growth and development, but also regulate seed size [10]. In rice, grain size is the main determinant of yield, and the spikelet hull is a key determinant of grain size [11].” For example. Move these sentences later in the paragraph, after mentioning the miRNAs, and modify to something like: “ In addition to these well-known pathways, transcription factors and plant hormones have been involved in the control of ovule and seed size. Moreover, the correct development of other plant structures is also key determinant of grain size and hence of crop yield, such as the spikelet hull in rice”. This is just an idea on how to improve this paragraph.
Response 3: Thanks for your advice. We had deleted “Transcription factors regulate cell growth in the ovule, resulting in seed growth. Plant hormones not only regulate plant growth and development, but also regulate seed size [10]. In rice, grain size is the main determinant of yield, and the spikelet hull is a key determinant of grain size [11].”, and added “In addition to these well-known pathways, transcription factors and plant hormones have been involved in the control of ovule and seed size. Moreover, the correct development of other plant structures is also key determinant of grain size and hence of crop yield, such as the spikelet hull in rice” in the revised manuscript.
-I would recommend one-two sentences at the beginning of the sections 2, 3, 4, and 5, explaining what these pathways are and their function at the cellular-biochemical level as well as during plant development, not only the role of individual genes in seed development. They have done a brief introduction to the MAP kinase pathway (section 5).
Response 4: Thanks for your advice. We had introducted the pathways at the beginning of the sections 2, 3, 4, 5 in the revised manuscript.
Sections 2: The IKU pathway mainly regulates early endosperm development and includes IKU1, IKU2, MINISEED3 (MINI3), and SHORT HYPOCOTYL UNDER BLUE1 (SHB1) genes.
Sections 3: Ubiquitin-proteasome system is the main pathway of protein degradation in cells, which including ubiquitin, ubiquitin activating enzyme E1, ubiquitin binding enzyme E2, ubiquitin ligase E3, 26S proteasome and ubiquitin dissociating enzyme.
Sections 4: G proteins are a class of signal transduction proteins that can bind with guanosine diphosphate (GDP) and possess GTPase activity. The signaling pathways involving G proteins are highly conserved transmembrane signaling mechanisms in both animals and plants. When cells transduce extracellular signals, they are first received by different types of G protein-coupled receptors (GPCRs), which act as receptors for various ligands (extracellular first messengers). Subsequently, the receptors are activated, further activating heterotrimeric G proteins on the inner side of the plasma membrane. The activated G proteins then go on to activate various effectors downstream, generating second messengers within the cells. This sequential signal transduction cascade regulates the growth and development processes of organisms
Sections 5: MAPK cascades are conserved and involved in plant signal transduction in eukaryotes. The MAPK cascades consist of three functional protein kinases: MAPKs, MAPK kinases (MAPKKs), and MAPK kinase kinases (MAPKKKs).
-Move sentence in line 183-184 (The OsMKKK10-OsMKK4-OsMAPK6 signaling pathway positively regulated rice grain size and weight) at the beginning of the paragraph. In this way, they first introduce the conclusion or message and elaborate it afterwards. If not, change the sentence to something like “In summary, data clearly indicate that OsMKKK10-OsMKK4-OsMAPK6 signaling pathway positively regulated rice grain size and weight”.
Response 5: Thanks for your advice. We had revised it as “In summary, data clearly indicate that OsMKKK10-OsMKK4-OsMAPK6 signaling pathway positively regulated rice grain size and weight”.
-Rewrite sentence in line 221-222 to “The Alfin-like (AL) family was a group of small plant-specific transcriptional factors. Rice mutants al7.1 and al11 have increased seed size” or “The Alfin-like (AL) family was a group of small plant-specific transcriptional factors, and null mutants of the rice OsAL7.1 and OsAL11 have increased seed size”.
Response 6: Thanks for your advice. We had revised it as “The Alfin-like (AL) family was a group of small plant-specific transcriptional factors, and null mutants of the rice OsAL7.1 and OsAL11 have increased seed size”.
-Revise the text to check for mutant/gene names. For example, in lane 263, “ZmARF12 mutant” is not correct; either use “zmarf12 mutant”, “maize arf12 mutant” or “mutant of ZmARF12”.
Response 7: Thanks for your advice. We had revised it as “mutant of ZmARF12”.
-What is the relationship among ARF12, GIF1, and GRF in paragraph lines 261-271? Unless it is justified, I would move the data for ARF12 to the auxin section 7.1, where they mention data for other ARFs, like wheat ARF25 and JcARF19.
Response 8: Thanks for your advice. We had moved the data for ZmARF12 to the auxin section 7.1.
-Add maize in the list of plants in lines 40-41, and remove the plant name (Zea mays) in line 295-296. Change sentence to “This article reviews the research results on the mechanisms that regulate seed size in plants such as Arabidopsis (Arabidopsis thaliana L.), wheat (Triticum aestivum L.), rice (Oryza sativa L.), maize (Zea mays L.), and other plants.
Response 9: Thanks for your advice. We had added “Zea mays L.” in line 40 and removed “Zea mays ” in line 295.
And we had changed the sentence to “This article reviews the research results on the mechanisms that regulate seed size in plants such as Arabidopsis (Arabidopsis thaliana L.), wheat (Triticum aestivum L.), rice (Oryza sativa L.), maize (Zea mays L.), and other plants.
-Please, revise the sentence in lines 340-342 “TaGW2-6A encoded a RING E3 ubiquitin-ligases with actively involved in GA perception and signal transduction and negatively regulated grain size [79].” First, the word “with” should be eliminated, but moreover it is not clear to me that a E3 ubiquitin-ligase is involved in GA perception. The author of reference #79 mentioned this in the abstract, but they do not refer to TaGW2-6A in particular. I would change to “TaGW2-6A encodes a RING E3 ubiquitin-ligases, which are involved in GA signal transduction, and negatively regulated grain size [79].
Response 10: Thanks for your advice. We had revised the sentence to “TaGW2-6A encodes a RING E3 ubiquitin-ligases, which are involved in GA signal transduction, and negatively regulated grain size.
-Line 375. CK is not a singular molecule (line 375); cytokinins (CKs) are a group of phytohormones. Change accordingly. Same for GAs (line 331).
Response 11: Thanks for your advice. We had changed it in the revised manuscript (marked in red).
-Line 461, what is the nature of the mOs07g46990 transgenic plants? Is it a knockout/knockdown mutant, or and overexpressor?
Response 12: Is it a knockdown mutant.
-Line 477, remove “and so on” and elaborate the idea. It is too informal.
Response 13: Thanks for your advice. We had revised in the revised manuscript.
-I would also recommend introducing Figure 1 in all sections (from section 2), as it may help to follow the text, not only at the end. It is the only figure and is only mentioned briefly at the end of the text.
Response 14: Thanks for your advice. We had revised in the revised manuscript.
-Figure 1, unless better explained, GA2ox8 would reduce GA levels, not affect directly seed size independently of the GAs. In addition, modify to GA2ox8 (not GA20x8), as it may induce some confusion to the GA20 oxidases. And very important, GAs negatively regulate DELLA protein stability, not as indicated in the figure. I recommend revising very carefully all Figure 1 to look for similar mistakes in other pathways.
Response 15: Thank you very much for your careful review, we had revised Figure 1 carefully.
Other text issues
-It seems that all italics of the text have been eliminated. Change to italics all plant names, gene names, and mutant names.
Response 16: We had changed to italics all plant names, gene names, and mutant names in the revised manuscript.
-Some corrections must also be done in the text. I have some examples. I recommend a complete revision and editing of the text.
For example:
-Line 46 Change to “IKU1 encodes a protein containing the plant-specific VQ motif”.
Response 17: We had changed it.
-Line 261, rewrite “a member transcription factor”. Something is missing in this sentence.
Response 18: We had deleted “a member transcription factor “ and changed it to “Auxin Response Factor 12 of maize (ZmARF12), which mediated the expression of auxin-responsive genes, regulated seed size development, as mutant of ZmARF12 exhibited larger seed size and higher grain weight”.
-Line 289, ARFs have already been spelled out before (in line 261). No need to do it again here.
Response 19: Yes, thank you very much for your careful review, and we had revised.
-Line 331, change to “GAs are important phytohormones that play a crucial role...”
Response 20: Yes, thank you very much for your careful review, and we had changed.
-Line 350, missing comma after coi1-8? Also missing comma after triple mutant in line 354, and after MYC4 in line 355. Again, add comma after pft1-3 in line 358. Revise all text.
Response 21: Thank you very much for your careful review, and we had revised.
-Line 388-391; Rephrase to “Treatment of thidiazuron (TDZ), a novel and efficient cytokinin, increased seed diameter and silique length, as well as the yield, since the embryo and cotyledon epidermal cells were obviously larger in TDZ-treated Brassica napus plants [89].
Response 22: Thanks for your advice. We had rephrased to “Treatment of thidiazuron (TDZ), a novel and efficient cytokinin, increased seed diameter and silique length, as well as the yield, since the embryo and cotyledon epidermal cells were obviously larger in TDZ-treated Brassica napus plants”.
-Line 395, change “Arabidopsis histidine kinases (AHKs) are receptors in the CK signaling pathway” to “Arabidopsis histidine kinases (AHKs) are CK receptors”, or “Arabidopsis histidine kinases (AHKs) are the receptors in the CK signaling pathway”.
Response 23: Thanks for your advice. We had changed “Arabidopsis histidine kinases (AHKs) are receptors in the CK signaling pathway” to “Arabidopsis histidine kinases (AHKs) are the receptors in the CK signaling pathway”.
-Line 413, change critical to important.
Response 24: Thanks for your advice. We had changed.
-Line 437, sentence should begin with lower case “miRNAs are a class…”. Same in line 451 for miR529a, and through the text.
Response 25: Thanks for your advice. We had changed.
-Line 437-239, rewrite “MiRNAs are a class of small noncoding RNAs that play key regulatory roles in plant seed development, and the Zma-miRNA169 family is highly expressed during maize seed development [99]. Overexpression of maize zma-miR169o increased seed size and weight, while inhibiting its expression had the opposite effect [99]”. Change to something like: “miRNAs are a class of small noncoding RNAs that play key regulatory roles in seed development. For example, overexpression of maize zma-miR169o increased seed size and weight, whereas inhibiting its expression had the opposite effect [99]”.
Response 26: Thanks for your advice. We had changed it to “miRNAs are a class of small noncoding RNAs that play key regulatory roles in seed development. For example, overexpression of maize zma-miR169o increased seed size and weight, whereas inhibiting its expression had the opposite effect”.
-Line 448-450, this sentence is quite confusing; rewrite. “both transgenic plants overexpressing (miR529a-OE) and suppressing (miR529a-MIMIC) the miR529a resulted in narrower grain and lower grain weight, miR529a-OE plants produced longer and slenderer grains, whereas miR529a-MIMIC plants produced shorter grains [101]”.
Response 27: Thanks for your advice. We had changed it to “Both transgenic plants overexpressing (miR529a-OE) and suppressing (miR529a-MIMIC) the miR529a resulted in narrower grain and lower grain weight, miR529a-OE plants produced longer and slenderer grains, whereas miR529a-MIMIC plants produced shorter grains”.
-Line 454-458, rewrite to indicate what exactly STTM159 is: “By using Short Tandem Target Mimic (STTM), Zhao et al. successfully suppressed the expression of mature miR159 in STTM159 transgenic plants, leading to increased expressions of its two target genes, OsGAMYB and OsGAMYBL1 (GAMYB-LIKE 1) [102]. Moreover, STTM159 plants were dwarfed with reduced organ size, stem diameter, length of flag leaf, main panicle, spikelet hulls and grain size [102].
These are only few examples, I did not go through all the text, so I recommend a deep proofreading/editing of the text.
Response 28: Thanks for your advice. We had changed it to “By using Short Tandem Target Mimic (STTM), Zhao et al. successfully suppressed the expression of mature miR159 in STTM159 transgenic plants, leading to increased expressions of its two target genes, OsGAMYB and OsGAMYBL1 (GAMYB-LIKE 1). Moreover, STTM159 plants were dwarfed with reduced organ size, stem diameter, length of flag leaf, main panicle, spikelet hulls and grain size”.
Comments on the Quality of English Language
Some corrections must also be done in the text. I have included some issues/suggestions in my review, but strongly recommend a complete revision and editing of the text.
Response 29: Thanks for your advice. The revised manuscript had been checked by a colleague fluent in English writing.